# On the Additional Information Provided by 3T-MRI ADC in Predicting Tumor Cellularity and Microscopic Behavior

**DOI:** 10.3390/cancers13205167

**Published:** 2021-10-15

**Authors:** Domiziana Santucci, Eliodoro Faiella, Alessandro Calabrese, Bruno Beomonte Zobel, Andrea Ascione, Bruna Cerbelli, Giulio Iannello, Paolo Soda, Carlo de Felice

**Affiliations:** 1Department of Radiology, University of Rome “Campus Bio-Medico”, Via Alvaro del Portillo 21, 00128 Rome, Italy; e.faiella@unicampus.it (E.F.); b.zobel@unicampus.it (B.B.Z.); 2Department of Radiology, University of Rome “Sapienza”, Viale del Policlinico 155, 00161 Rome, Italy; alessandro.calabrese.92@gmail.com (A.C.); c.df@uniroma1.it (C.d.F.); 3Department of Radiological, Oncological and Pathological Science, University of Rome “Sapienza”, Viale del Policlinico 155, 00161 Rome, Italy; andrea.ascione@uniroma1.it (A.A.); bruna.cerbelli@uniroma1.it (B.C.); 4Unit of Computer Systems and Bioinformatics, Department of Engineering, University of Rome “Campus Bio-Medico”, Via Alvaro del Portillo 21, 00128 Rome, Italy; g.iannello@unicampus.it (G.I.); p.soda@unicampus.it (P.S.)

**Keywords:** breast cancer, 3T-MRI, apparent diffusion coefficient (ADC), cellularity, grading, Ki-67 index, tumor aggressiveness

## Abstract

**Simple Summary:**

Breast cancer is the most common cancer in women worldwide. Increasing knowledge of the microscopic behavior of tumors has allowed for personalized and increasingly effective therapies. Biopsy is the first step in the histological evaluation of breast cancer. However, biopsy may only be partially representative of the entire tumor. The only currently recognized independent histological prognostic factor is grading, an expression of replicative cellular behavior. However, other factors such as the Ki-67 proliferation index and tumor cellularity provide additional information on tumor aggressiveness. MRI is the imaging technique routinely used in the loco-regional tumor staging phase. Currently, the MRI protocol includes DWI sequences. DWI is an expression of the restriction of water molecules and can be quantified through ADC values. In our work, entitled “On the additional information provided by 3T-MRI ADC in predicting tumor cellularity and microscopic behavior”, we aim to demonstrate how ADC can significantly correlate with these histological factors, in particular with the cellularity obtained in the definitive histological sample compared to the biopsy sample; ADC values may therefore offer a valuable support for biological evaluation in the pre-surgical phase.

**Abstract:**

Background: to evaluate whether Apparent Diffusion Coefficient (ADC) values of invasive breast cancer, provided by 3T Diffusion Weighted-Images (DWI), may represent a non-invasive predictor of pathophysiologic tumor aggressiveness. Methods: 100 Patients with histologically proven invasive breast cancers who underwent a 3T-MRI examination were included in the study. All MRI examinations included dynamic contrast-enhanced and DWI/ADC sequences. ADC value were calculated for each lesion. Tumor grade was determined according to the Nottingham Grading System, and immuno-histochemical analysis was performed to assess molecular receptors, cellularity rate, on both biopsy and surgical specimens, and proliferation rate (Ki-67 index). Spearman’s Rho test was used to correlate ADC values with histological (grading, Ki-67 index and cellularity) and MRI features. ADC values were compared among the different grading (G1, G2, G3), Ki-67 (<20% and >20%) and cellularity groups (<50%, 50–70% and >70%), using Mann–Whitney and Kruskal-Wallis tests. ROC curves were performed to demonstrate the accuracy of the ADC values in predicting the grading, Ki-67 index and cellularity groups. Results: ADC values correlated significantly with grading, ER receptor status, Ki-67 index and cellularity rates. ADC values were significantly higher for G1 compared with G2 and for G1 compared with G3 and for Ki-67 < 20% than Ki-67 > 20%. The Kruskal-Wallis test showed that ADC values were significantly different among the three grading groups, the three biopsy cellularity groups and the three surgical cellularity groups. The best ROC curves were obtained for the G3 group (AUC of 0.720), for G2 + G3 (AUC of 0.835), for Ki-67 > 20% (AUC of 0.679) and for surgical cellularity rate > 70% (AUC of 0.805). Conclusions: 3T-DWI ADC is a direct predictor of cellular aggressiveness and proliferation in invasive breast carcinoma, and can be used as a supporting non-invasive factor to characterize macroscopic lesion behavior especially before surgery.

## 1. Introduction

The highly heterogeneous nature of breast cancer is a consequence of the different histological behaviors of the tumor and the different molecular receptor structures. On the basis of this heterogeneity, an improvement in disease-specific survival rates was observed through increasingly personalized treatment.

The variable prognostic factors include tumor size, lymph node status, grade, histological type, and molecular features, such as expression of ER/PgR, HER2, and proliferation markers (e.g., Ki-67), which are used to divide BCs into the molecular-specific subtypes with different diagnostic and therapeutic pathway, risk of recurrence and response to treatment [1,2].

Molecular expression, related to the presence or absence of receptors on cell membranes (ER, PgR, etc.), is used to guide the choice of treatment. On the other hand, the cellular microenvironment is crucial to define the aggressiveness of breast cancer and can be quantified by cell proliferation and intrinsic cellular changes, such as mitotic count and degree of pleomorphism.

Ki-67 proliferation index is, presently, the most widely used marker to determine the degree of proliferation of human cancer cells, regulating cell cycle progression in human cells, and it is counted among the main prognostic factors [3]. Histological grade, classified using the Nottingham Grading System (NGS), represents a histological subdivision according to cellular differentiation. NGS is calculated by scoring three morphological features on a scale of 1 to 3: degree of tubule formation, nuclear pleomorphism and mitotic count [4]. While histological type can be a useful predictor on its own, more than 60% of breast cancers are non-special type invasive ductal carcinomas (NSTs). This makes histological grade a relatively better predictor when considered independently [5]. Grade is relatively simple and inexpensive to obtain, and several studies have shown that it can predict tumor behavior better than other prognostic factors alone [6,7,8,9].

Tumor cellularity is defined by the proportion of tumor cells in the tumor bed: this assessment is performed by a pathologist who estimates the local cellularity of the specimen by comparing the area containing the tumor with the reference standard. Being a semi-quantitative assessment, it is, therefore, subject to inter-rater variability [10].

At the time of this study, cancer cellularity is not employed as a method of further sub-differentiation of breast cancer and its role is reserved in assessing tumor response to neoadjuvant therapy, according to the Sinn, Sataloff, and Miller-Payne Pinder methods of evaluating response to treatment [11,12,13,14,15,16,17,18,19].

The current limit is due to the different cellularity score obtained from biopsy and definitive surgical specimens. The former, in fact, could be representative of a single portion and not of the whole tumor, distorting the final evaluation.

Among imaging techniques, the MRI examination is increasingly used in the staging phase of breast cancer, and its routine protocol includes diffusion weighted imaging (DWI): DWI improves the specificity of post-contrast-MRI, which is considered the gold standard for tumor detection. DWI restriction is proportional to the degree of movement of water molecules and can be quantified by the apparent diffusion coefficient (ADC) value. In ADC maps, tissues with high cellularity showed lower ADC values [20,21]. Recent literature has demonstrated an inverse correlation between ADC values and histological grade of the tumor, proving its usefulness in identifying high-grade invasive breast cancer prior to the surgery [22,23].

The aim of this study is to define whether ADC values, obtained by 3T staging-MRI, vary with Ki-67 expression, tumor grade and cellularity, the latter obtained both after biopsy and after surgery, in order to identify whether ADC might represent a non-invasive predictor of aggressive cellular definitive pathophysiology.

## 2. Materials and Methods

### 2.1. Study Population

In this study, all breast cancer MRI examinations performed at our Department of Radiological Sciences for local staging from January-2010 to September-2019 were retrospectively reviewed. A total of 100 patients with histologically proven invasive BC lesions were enrolled.

The following inclusion criteria were considered: staging 3T-MRI examination, performed after biopsy and before surgery; presence of DCE-MRI, T2-WI and DWI sequences; ADC evaluation of the main lesion for each exam; histopathological diagnosis confirming invasive BC; complete histological analysis including molecular receptor assessment (estrogen receptor ER, progesterone receptor PgR; epidermal growth factor receptor HER2), Ki-67 index, and calculation of cellularity, both at biopsy and on the operative specimen.

Exclusion criteria were: presence of breast implants, post-chemotherapy follow-up examinations, neo-adjuvant chemotherapy and images that were not of good diagnostic quality.

Patients clinical data (age, menopausal state, familiarity, hormone therapy), tumor MRI features (stadiation, localization, margins, kinetic curves, size) and histological features (histological type, grading, ER, PgR, HER2, Ki-67 index, cellularity rate) were collected.

Institutional Review Board approval was not required because this is a retrospective observational study, and only existing information collected from human participants was used and there are no identifiers linking individuals to the data/samples.

All methods and procedures were in accordance with institutional and research committee ethical standards and the 1964 Declaration of Helsinki and its subsequent amendments or comparable ethical standards.

### 2.2. MRI Examination

All MRI exams were performed on a 3T magnet (Discovery 750; GE Healthcare, Milwaukee, WI, USA). Patients were positioned prone and a dedicated eight-channel breast coil (8US TORSOPA) was employed. Three orthogonal localizer sequences were performed, then the following protocol was acquired:T2-weighted axial single-shot fast spin echo sequence with fat suppression (DIXON) (TR/TE 3500–5200/120–135 ms, matrix 352 × 224, FoV 370 × 370, NEX 1, slice thickness 3.5 mm).Diffusion weighted axial single-shot echo-planar sequence with fat suppression (TR/TE 2700/58 ms, matrix 100 × 120, FOV 360 × 360, NEX 6, slice thickness 5 mm) with b values of 0, 500 and 1000 s/mm^2^.T1-weighted axial 3D dynamic gradient echo sequence with fat suppression (VIBRANT) (TR/TE 6.6/4.3 ms, flip angle 10°, matrix 512 × 256, NEX 1, slice thickness 2.4 mm), before and five times after intravenous contrast medium injection.

Current guidelines suggest at least three time points to measure during the post-contrast-phase: one before the administration of contrast medium, one approximately 2 min later to capture the peak, one in the late phase. This allows us to evaluate whether a lesion continues to enhance or is characterized by contrast agent wash-out. At least two measurements after contrast medium administration are recommended, even if the optimal number of repetitions is unknown. In our center, we usually perform five acquisitions after contrast medium administration ensuring obtaining a specific signal intensity curve time without penalizing the duration of the examination.

Gadobenate-dimeglumine (Multihance^®^; Bracco Imaging, Milan, Italy) was administered as contrast agent (concentration of 0.2 mmol/kg; rate of 2 mL/s) followed by injection of 15 mL of saline. In post-processing, subtracted images were automatically produced from the images after contrast medium administration for a more accurate tumor analysis.

The entire exam for each patient was transferred to a workstation (Advantage Windows Workstation 4.4; GE Medical System, Milwaukee, WI, USA) for post-processing analysis. For a quantitative analysis, ADC values were calculated according to the following equation:ADC = −(1/b)ln (S0/S1)
where b is the diffusion factor, S1 is the attenuated signal (b-value of 1000 s/mm^2^) and S0 is the full spin echo signal without diffusion gradient (b-value of 0 s/mm^2^), as reported in the literature [24].

DCE sequences were considered to be reference images for tumor detection and lesion characterization. The largest lesion was considered to be the index lesion and included in the statistical analysis. The greatest axial diameter was submitted to statistical analysis. On the basis of their morphology, lesions were classified into mass tumors with regular, lobulated, irregular and spiculated margins, or non-mass tumors.

For each index lesion, a signal intensity-to-time curve (SI/T) was automatically generated by placing a region of interest (ROI) within the lesion on a subjectively recognized area of maximal contrast enhancement and evaluating all five of the acquired DCE series. The kinetics curves were classified as I (progressive wash-in), II (plateau) or III (rapid wash-out), in accordance to the BIRADS guidelines.

For qualitative analysis, DWI sequences were subsequently evaluated and the lesion was simply considered to be visible (characterized by diffusion restriction) or non-visible (without any diffusion restriction). For quantitative analysis, the ADC value of the index lesion was calculated by superimposing the subtracted images on the ADC map. The ROI was circular, measuring 3–6 mm, and was manually drawn on the slice where the lesion reached its greatest diameter. Then, the ADC value was generated and stored automatically. ADC measurements were performed only on the enhanced solid portion to avoid areas of T2 shine-through, i.e., the necrotic core of the tumor. All ADC values were retrospectively measured by a radiologist with more than 10-years of experience with breast MRI, as they were not originally included in the reports.

### 2.3. Histologic Characteristics

All breast lesions were characterized on the histological specimen obtained by core biopsy and on the histological definitive sample after surgery, by two pathologists. Histological diagnosis was performed according to WHO classification. In the cases with lobular histotype where assessing cellularity was particularly troublesome, the count was performed on sections stained with immunohistochemistry for cytokeratin (CKAE1AE3).

The histopathological grade was evaluated according to NGS considering the tubule formation, the pleomorphism and the mitotic count through a scoring system. The total score ranges from 3 to 9: 3–5 corresponds to grade 1 (G1), 6 or 7 to grade 2 (G2) and 8 or 9 to grade 3 (G3).

Immunohistochemical (IHC) analysis was performed to evaluate molecular receptors status (ER, PgR, and HER2) and to calculate Ki-67 index. Evaluation of ER and PgR status was performed by IHC using Dako monoclonal antibody, 1:100 dilution. The monoclonal antibody Mib-1 (1:200 dilution; Dako, Glostrup, Denmark) was used to assess the Ki-67 index, which was reported as the percentage of immune-reactive cells out of 2000 tumor cells in randomly selected high-power fields surrounding the tumor core. HER2 status was re-evaluated using the Hercep test (Dako, Glostrup, Denmark), following published guidelines [16]. Samples that gave an equivocal IHC result were subjected to fluorescence in situ hybridization (FISH) analysis. A ratio of HER2 gene signals to chromosome 17 signals greater than 2.2 was used as a cut-off value to define HER2 gene amplification.

ER and PgR status were considered to be positive if the expression was ≥1% and negative if the expression was <1%. HER2 expression was classified as 0, 1+, 2+ or 3+; only tumors reaching a score of 3+ were considered to be HER2-positive. The lesions were divided into four groups based on the rate of Ki-67 index: <10%, between 10% and 14%, between 14% and 20%, and >20%, respectively. Patients were further grouped according to Ki-67 expression into Ki-67 index <20% and >20%, considering the first group as unequivocally negative and the second as unequivocally positive. Cancer cellularity was assessed semi-quantitatively on the biopsy specimen and the surgical specimen by estimating the percentage of the tumor area covered by neoplastic cells. When more than one tumor bed was identified, cellularity was calculated as an average of each area’s cellularity, weighted for its approximate size. Foci of necrosis and in situ carcinoma were excluded from the assessment. Subdivision was made into groups as follows: <50%, between 50% and 70%, >70%, respectively.

### 2.4. Statistical Analysis

The ADC was treated as a continuous dependent variable, whereas Ki-67 expression, tumor grade and the cellularity rate, were considered to be independent variables. All the analyzed variables did not follow a normal distribution, and non-parametric tests were used for statistical computations.

Spearman’s Rho correlation test was used to correlate ADC values with histological features (histotype, class, ER, PgR, HER2, grading, Ki-67, biopsy cellularity and surgical cellularity) and with other MRI features (tumor size, kinetic curves, and margins).

To detect significant differences in ADC values among the grading groups, Ki-67 groups and biopsy and cellularity groups, both the Wilcoxon–Mann–Whitney U test (two groups comparison: G1 vs. G2, G2 vs. G3 and G1 vs. G3; Ki-67 < 20% vs. Ki-67 > 20%; cellularity < 50 vs. cellularity 50–70%, cellularity 50–70% vs. cellularity > 70%, cellularity < 50 vs. cellularity > 70%) and Kruskal–Wallis H test (multiple-groups comparisons: G1 vs. G2 vs. G3; cellularity < 50 vs. cellularity 50–70% vs. cellularity > 70%) were carried out.

A ROC curve was performed to demonstrate the accuracy of ADC values in predicting the most aggressive patterns: G3 class alone, G2 + G3 classes together, the Ki-67 > 20% group and the surgical cellularity >70% group. An Area Under the Curve (AUC) > 9 indicated an excellent test, between 8 and 9 a good test, between 7 and 8 a fair test, between 6 and 7 a poor test and <6 a worthless test.

Statistical significance was set at *p* < 0.05. All data analyses were processed using SPSS (IBM Statistical Software Program, IBM, Armonk, NY, USA), version 25.0.

## 3. Results

A total of 100 histologically proven invasive breast carcinomas were included in the study, and their MRI examinations were retrospectively reviewed. The mean age was 54.72 years (range 38–83 years); 54 patients were postmenopausal, 46 premenopausal; 10 had undergone hormone therapy during their lifetime; 24 patients had a relative with a history of breast cancer, and 12 had two or more affected relatives. Descriptive statistics for tumor MRI characteristics, which are reported for each classification group along with relative frequencies, and Spearman’s Rho test results are summarized in Table 1.

The histopathological and immunohistochemical results for all lesions and classification groups, with the results of Spearman’s Rho test, are shown in Table 2.

The mean ADC value of the lesions detected at DWI was 1.09 × 10^−3^ mm^2^/s (range: 0.7–1.5 × 10^−3^ mm^2^/s). No statistical significance was found by correlating ADC values with kinetic curves and tumor size; a significant correlation was found between ADC values and tumor margins (*p* = 0.032). No statistical significance was found between ADC values and histological type, PgR and HER2 receptor status (*p* > 0.05). However, ADC correlates significantly with grading, ER receptor status and Ki-67 index (*p* < 0.05). A high correlation was found between ADC values and cellularity rate, both biopsy (*p* < 0.01) and surgical (*p* << 0.001) (Figure 1, Figure 2 and Figure 3).

The Wilcoxon-Mann–Whitney test showed significant different ADC values when the following grading groups were compared individually, with a p value of 0.009 for G1 vs. G2 and a p value of 0.001 for G1 vs. G3, but no significant difference was found for G2 vs. G3. ADC values were significantly higher for Ki-67 < 20 % than Ki-67 > 20%, with a *p* value < 0.03. There was a significant difference in ADC values when the biopsy cellularity groups were compared individually, with a p value of 0.014 for <50% vs. >70%. No significant difference was found for the other classes (<50% vs. 50–70% and 50–70% vs. >70%). ADC values were also statistically different for the comparison of the individual surgical cellularity groups, with a *p* value << 0.001 for <50% vs. >50–70%, whereas *p* was 0.07 for 50–70% vs. >70% and *p* << 0.001 for <50% vs. >70%.

The Kruskal-Wallis test showed that ADC values were significantly different among the three grading groups (*p* = 0.009), the three biopsy cellularity groups (*p* = 0.21) and, most strongly, the three surgical cellularity groups (*p* << 0.001).

Using ADC values, the prediction for G3 corresponded to an AUC of 0.720, whereas it corresponded to a score of 0.835 if G2 and G3 were grouped (Figure 4 and Figure 5 respectively). The AUC for Ki-67 < 20% corresponded to 0.679 (Figure 6). The AUC for surgical cellularity > 70% was 0.805 (Figure 7).

## 4. Discussion

ADC values and their correlation with malignant lesions have been extensively studied in several papers, which have shown that ADC values of infiltrating lesions are statistically lower than those of in situ tumors, and that similarly ADC values of in situ tumors are lower than those of B3 lesions and these in turn of benign lesions [25]. Since ADC values are an expression of tumor cellularity, perfusion and angiogenesis, in the context of malignant tumor lesions they can be used to assess the degree of malignancy and aggressiveness, being inversely related to tumor grade and Ki-67 index, as already demonstrated by literature [26,27,28,29,30,31].

Therefore, while DWI cannot be strictly considered to be an expression of the presence or absence of receptors on cell membranes, it can be representative of the microscopic assessment of morphological and cytological features of tumor cells: those include the degree of tubule formation, nuclear pleomorphism, and mitotic count, all of which are included in the grading assessment.

The histological tumor grade is one of the most effective and practical prognostic factors in breast cancer [32]. The morphological characteristics of the tumor mass correlate with its biological and clinical behavior and thus can accurately predict the response to therapy and the patient prognosis. The prognostic significance of the Nottingham Grade System was first demonstrated in 1991 [4], providing important tumor information with reproducible results. NGS has been shown to be a simple, easily reproducible, and relatively cost-effective method for predicting tumor evolution and prognosis. For this reason, it has not been completely superseded by newer molecular tests, which have cost limitations, are not as widespread as grade analysis [9], and may lead to misclassification in the presence of rare histologic subtypes (e.g., neuroendocrine tumors) [33].

Our aim is to test in our sample whether ADC, obtained from 3T MRI examination, can be used as a predictable non-invasive index of tumor aggressiveness and whether it can summarize the macroscopic biological behavior of the tumor, considering not only grading and Ki-67, but also exploring the role of ADC values in cellularity prediction.

The results present in the literature concerning the correlation between ADC and grading are highly variable between studies. In particular, some papers did not find a significant association between ADC values and histological grade of breast cancer [34,35] and there is currently no consensus on the b values to determine ADC values, and no cut-off has yet been proved to predict pre-biopsy tumor grade.

In our study, a statistically significant inverse correlation between ADC value and tumor grading was demonstrated. G1 tumors presented significantly higher ADC values compared with G2 tumors, and also G2 compared with G3 tumors. Our results are in agreement with Yuan et al., who found significantly lower ADC values in the high pathological grade class compared to the low pathological grade class (*p* = 0.001), with a median value of 0.864 × 10^−3^ mm^2^/s for G2 + G3 lesions vs. 0.946 × 10^−3^ mm^2^/s for G1 lesions [27].

The relationship between ADC values and histological grade was also evaluated by ROC analysis. The ROC curve showed that the evaluation of ADC values was a fair test when considering the accuracy in predicting tumors in the G3 group (AUC = 0.720); interestingly, an AUC = 0.835 was obtained if G2 and G3 were combined. When compared to the study by Kızıldağ Yırgın et al. [29], our AUC was smaller considering the G3 class alone (0.720 vs. 0.875), but very similar when G2 and G3 were combined together (0.835 vs. 0.840). These results, expressed by the ROC curves, reflect the difficulty of distinguishing G2 tumors from the other two classes of lesions and in particular from G3 tumors, as is the case in anatomo-pathological evaluations. ADC is nothing more than an ex-vivo expression of the histological behavior of the lesion: to obtain a better predictive value of ADC, a better histo-pathological stratification between G2 and G3 classes is needed. While it is easier to identify the G1 class, the G2 definition is still controversial.

Ki-67 is a protein that is not expressed in the G0 phase of the cell cycle, and is therefore closely linked to tumor proliferation and consequently to tumor aggressiveness and cellularity. As cellularity increases, there is a reduction in the free diffusion of water molecules, which in turn corresponds to a reduction in ADC values. Ki-67 has also already been shown to be a good indicator of response to neoadjuvant chemotherapy and is a useful predictor of pCR [36]. In the field of MRI, changes in ADC values can be a non-invasive alternative to biopsy to estimate the effect of Ki-67-positive BC chemotherapy [37].

Our results regarding the correlation between ADC values and Ki-67 are in line with what has been basically demonstrated in the literature, where an inverse correlation between ADC and Ki-67 index has already been demonstrated [38,39,40,41]. Ki-67 positive tumors have significantly lower ADC values than Ki-67 negative tumors. In our study, we obtained an AUC of 0.679 in differentiating between Ki-67 positive BCs, which is similar to the AUC estimated by Shen et al. (0.683) even if they considered Ki-67 status as positive when Ki-67 index was greater than 14% [40]. Our AUC was significantly lower than Mori et al., who obtained an AUC of 0.81: this phenomenon is probably explained by the lower heterogeneity of their study sample, in which the included lesions were composed solely of luminal-type invasive breast cancers NOS [41].

Cellularity is presently assessed in the breast cancer specimen after surgery, reserving the assessment of cellularity biopsy only for patients destined for neo-adjuvant therapy. Some studies have shown that cellularity rates are already evident on MRI exams, and ADC values may help in distinguishing between benign and malignant lesions and predict in the early stages the patients who may respond to the eventual neoadjuvant treatment. Cytotoxic effects are, in fact, responsible for a reduction in tumor cellularity, resulting in reduced signal in DWI-weighted sequences and increased ADC values [42,43]. However, the biopsy cellularity cannot be representative of the real cellularity of the whole tumor, coming from a tumor portion sample. Ahn, S et al. and Reisenbichler, E et al. found no correlation between cellularity and survival, while other studies demonstrated a correlation between stroma-rich tumors and an increased risk of relapse and decreased survival, especially in triple-negative BC [11,12,13,14]. In the present work, we aimed to demonstrate an inverse correlation between ADC and tumor cellularity, and in particular the ability to more accurately predict the cellularity of the definitive sample compared with the biopsy sample, providing an additional decision support tool in the pre-surgical phase.

To our knowledge, there are no articles aiming to identify a non-invasive prognostic factor that can support the assessment of the cellularity of the biopsy specimen and, also, in specific cases, replace this assessment, giving indirect information on the histological aggressiveness of the tumor. We successfully demonstrated an inverse correlation between the pathophysiological phenomenon of cancer cellularity and the change in cellularity as shown in DWI sequences. The relative ROC curve showed that the evaluation of ADC values was a fair test when considering the accuracy in predicting tumors in the surgical cellularity >70% group (AUC = 0.805). These findings are in agreement with the results of Hatakenaka et al. [36] and Matsubayashi et al. [37], which, in contrast to us, studied cellularity in the biopsy specimen and in the neo-adjuvant setting.

The major contribution of our work lies in demonstrating that ADC values predict main macroscopic cell growth and replication factors in a highly accurate manner. In particular, our results reinforce the current literature regarding grading and Ki-i67, and introduce a new chapter regarding cellularity. There are no works that compare cellularity in the biopsy and in the surgical phases, therefore this work needs further validation. The ability of the ADC to predict surgical cellularity before surgery, during the MRI staging phase, certainly appears promising, offering an additional quantitative tool for the evaluation of the entire tumor, assisting the information obtained from the biopsy, which represents only tumor portions.

The main limitation of this preliminary study is that we did not perform a comparison of ADC values with biopsy/surgical cellularity values before and after treatment to define whether there is a correlation even during treatment. Additionally, another limitation is the presence of both IDCs and ILCs in the study population: the significantly lower number of ILCs may have contributed to bias in the statistics.

## 5. Conclusions

Our results indicate a significant correlation between ADC values detected on 3T-MRI in invasive breast cancers, biopsy and surgical cellularity, tumor histologic grade, and Ki-67 index. A higher ADC value corresponds to a lower cellularity rate, which in subsequent follow ups may be associated with a better response to treatment. Tumors with a lower histologic grade, corresponding to fewer mitoses, are characterized by a larger extracellular volume, corresponding to higher ADC values; vice versa for tumors with a high histologic grade. ADC values may predict histologic grade, but further studies are needed to evaluate the value of ADC as a prognostic factor to predict tumor behavior. In the future, ADC values could be taken together with DCE curves to achieve a better understanding of invasive breast cancers in an attempt to predict their evolution, prognosis, and response to therapy.

## Figures and Tables

**Figure 1 cancers-13-05167-f001:**
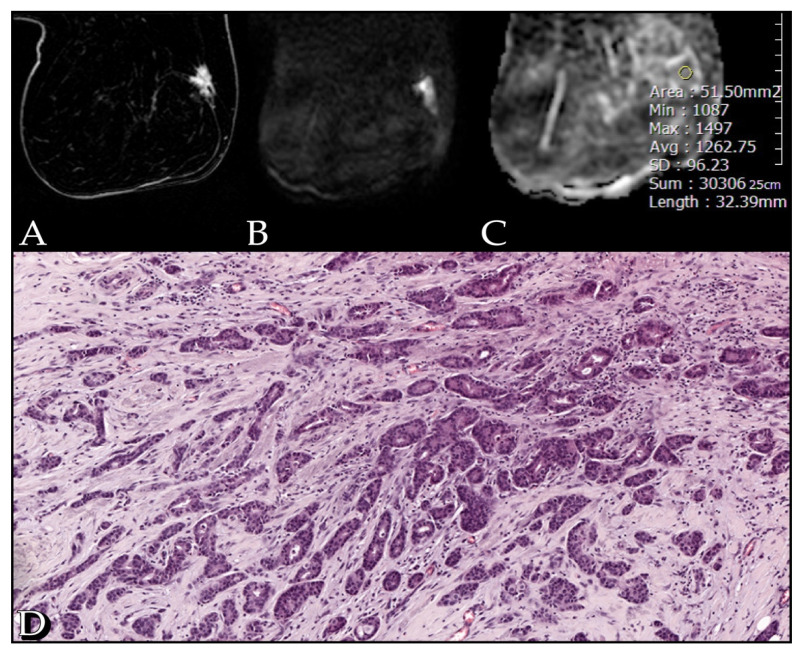
Case of a 51-year-old woman with a G1 Luminal A invasive ductal carcinoma, Ki67 of 12%. (**A**) The post-contrast image shows an irregular spiculated enhancing lesion in the outer quadrants of the right breast. (**B**) The diffusion weighted image (b-value of 1000 s/mm^2^) shows high restriction signal in the tumor region. (**C**) ADC map with an ADC value of 1.26 mm^2^/s. (**D**) Section from surgical specimen shows 40% cellularity (10× HE).

**Figure 2 cancers-13-05167-f002:**
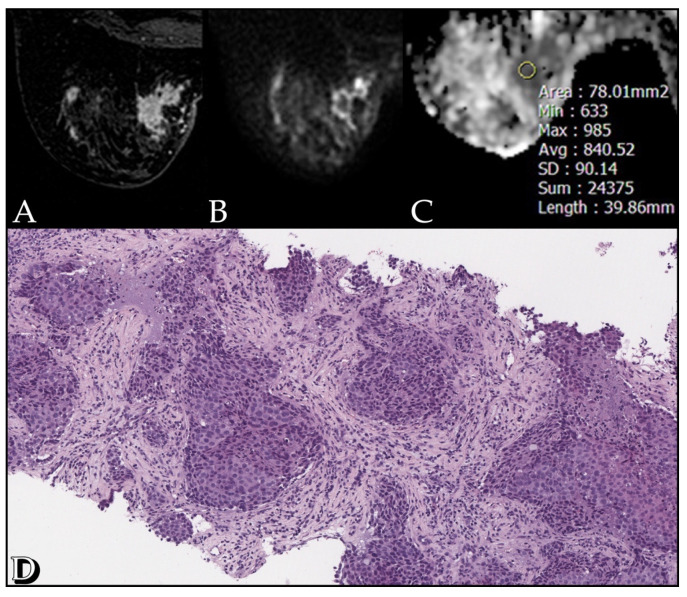
Case of a 51-year-old woman with a G2 Luminal A invasive ductal carcinoma, Ki67 of 60%. (**A**) The post-contrast image shows an irregular enhancing lesion in the inner inferior quadrant of the left breast. (**B**) The diffusion weighted image (b-value of 1000 s/mm^2^) shows high restriction signal in the tumor region. (**C**) ADC map with an ADC value of 0.84 mm^2^/s. (**D**) Section from surgical specimen shows 60% cellularity (10× HE).

**Figure 3 cancers-13-05167-f003:**
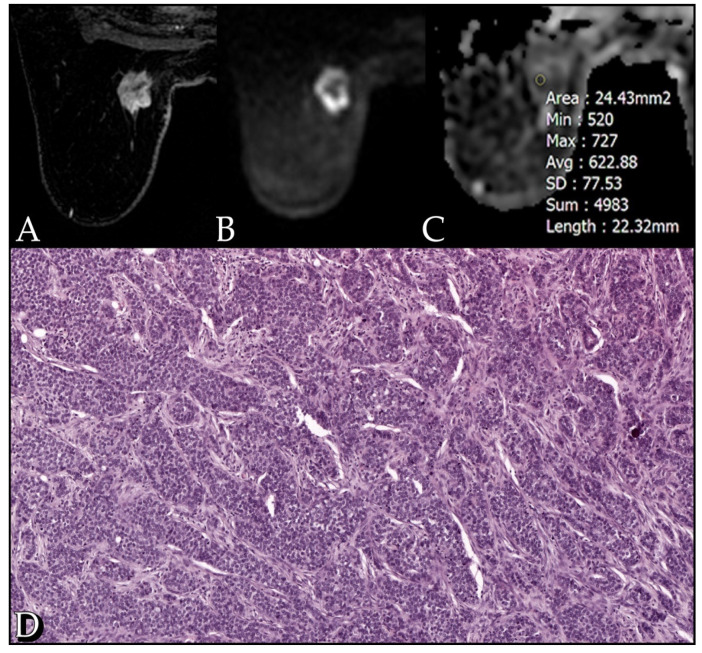
Case of a 72-year-old woman with a G3 Luminal B invasive lobular carcinoma. (**A**) The post-contrast image shows an irregular heterogeneously enhancing lesion in the inner upper quadrant of the left breast. (**B**) The diffusion weighted image (b-value of 1000 s/mm^2^) shows high restriction signal in the tumor region. (**C**) ADC map with an ADC value of 0.62 mm^2^/s. (**D**) Section from surgical specimen shows 90% cellularity (10× HE).

**Figure 4 cancers-13-05167-f004:**
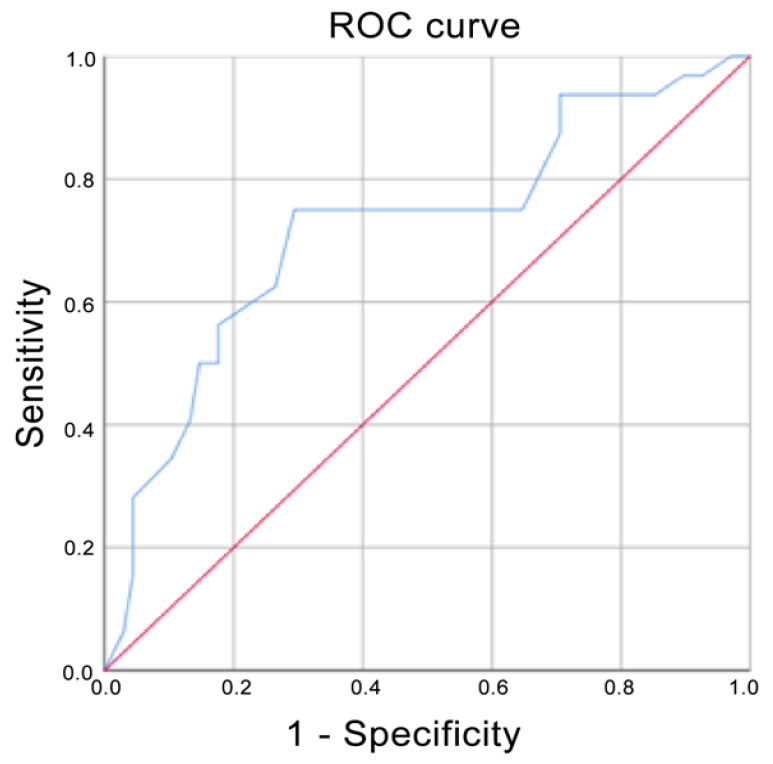
Receiver operating characteristic (ROC) curve when G3 was compared with G1 and G2. Area under the curve (AUC) = 0.720.

**Figure 5 cancers-13-05167-f005:**
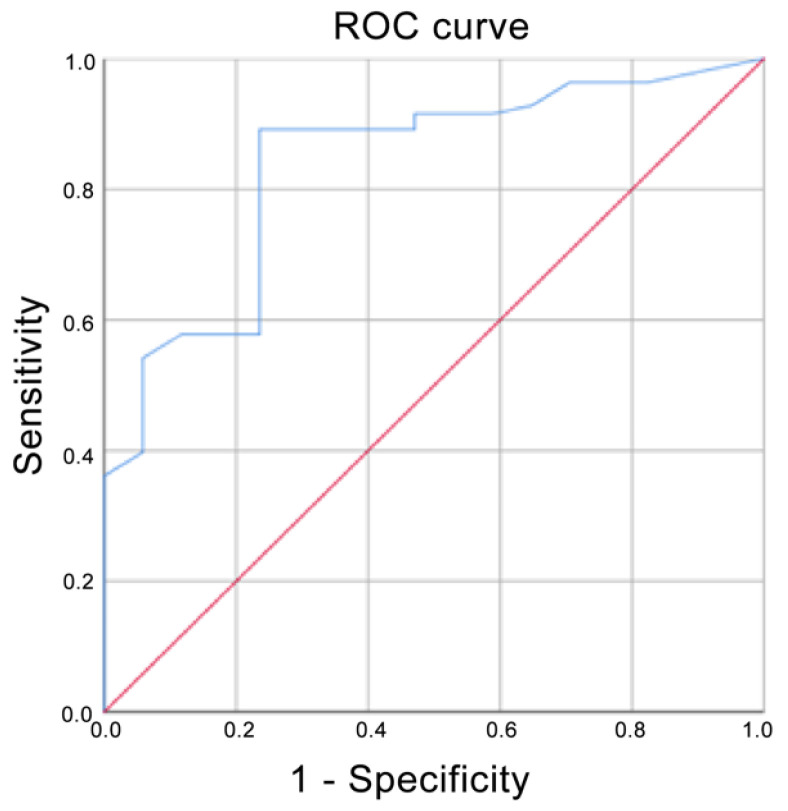
Receiver operating characteristic (ROC) curve when G2 and G3 were compared with G1. Area under the curve (AUC) = 0.835.

**Figure 6 cancers-13-05167-f006:**
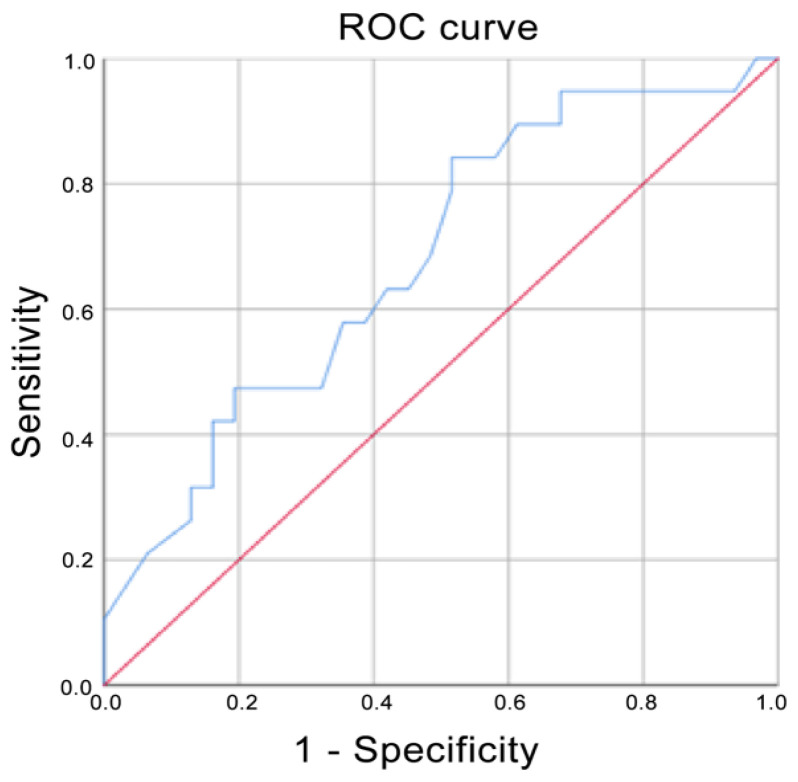
Receiver operating characteristic (ROC) curve when Ki-67 Index > 20% was compared with Ki-67 Index < 20%. Area under the curve (AUC) = 0.679.

**Figure 7 cancers-13-05167-f007:**
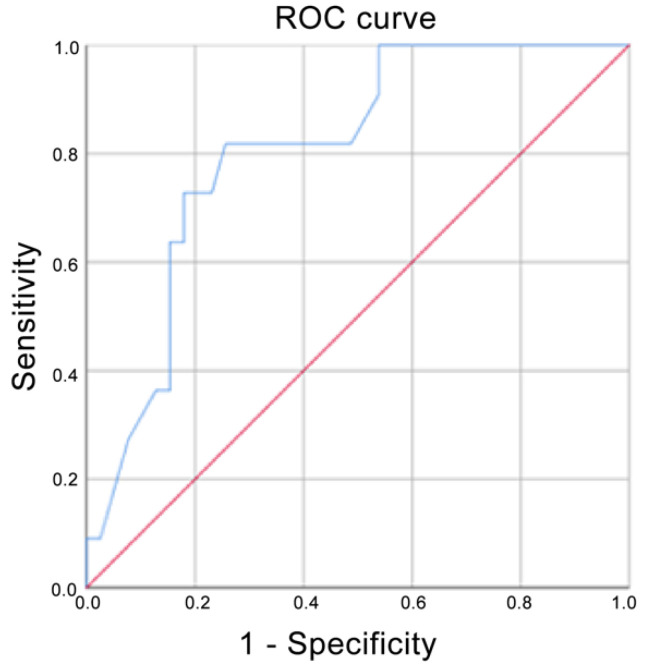
Receiver operating characteristic (ROC) curve when surgical cellularity rate > 70% was compared with surgical cellularity rate > 50 and surgical cellularity rate 50–70%. Area under the curve (AUC) = 0.805.

**Table 1 cancers-13-05167-t001:** Description of the extracted MRI characteristics. * indicates statistical significance (*p* < 0.05).

Variation			Grade				Surgical Cellularity Rate				Ki-67 Index			*p* Value
			1	2	3	Total	<50	50–70%	>70%	Total	<20%	>20%	Total	
Kinetic Curve	I	*n*	4	18	2	24	8	9	9	26	17	9	26	0.300
		%	4.0%	18.0%	2.0%	24.0%	8.0%	9.0%	9.0%	26.0%	17.0%	9.0%	26.0%	
	II	*n*	5	24	10	39	11	25	4	40	23	17	40	
		%	5.0%	24.0%	10.0%	39.0%	11.0%	25.0%	4.0%	40.0%	23.0%	17.0%	40.0%	
	III	*n*	3	19	14	36	9	13	12	34	20	14	34	
		%	3.0%	19.0%	14.0%	36.0%	9.0%	13.0%	12.0%	34.0%	20.0%	14.0%	34.0%	
Margins	Regular	*n*	0	7	2	9	2	2	4	8	5	3	8	0.032 *
		%	0.0%	7.0%	2.0%	9.0%	2.0%	2.0%	4.0%	8.0%	5.0%	3.0%	8.0%	
	Irregular	*n*	7	28	14	49	12	28	12	52	30	22	52	
		%	7.0%	28.0%	14.0%	49.0%	12.0%	28.0%	12.0%	52.0%	30.0%	22.0%	52.0%	
	Lobulated	*n*	4	12	3	19	7	7	4	18	12	6	18	
		%	4.0%	12.0%	3.0%	19.0%	7.0%	7.0%	4.0%	18.0%	12.0%	6.0%	18.0%	
	Spiculated	*n*	1	7	6	14	4	8	2	14	8	6	14	
		%	1.0%	7.0%	6.0%	14.0%	4.0%	8.0%	2.0%	14.0%	8.0%	6.0%	14.0%	
	Non-mass	*n*	0	8	1	9	3	2	3	8	5	3	8	
		%	0.0%	8.0%	1.0%	9.0%	3.0%	2.0%	3.0%	8.0%	5.0%	3.0%	8.0%	
Size	Mean 19.46 mm		19.67	19.00	20.07		17.32	19.87	21.08		20.38	18.08		0.560

**Table 2 cancers-13-05167-t002:** Description of the extracted histologic characteristics. * indicates statistical significance (*p* < 0.05).

Variation			Grade				Surgical Cellularity Rate				Ki-67 Index			*p* Value
			1	2	3	Total	<50	50–70%	>70%	Total	<20%	>20%	Total	
Histology	IDC	*n*	12	34	24	70	17	35	18	70	40	30	70	0.182
		%	12.0%	34.0%	24.0%	70.0%	17.0%	35.0%	18.0%	70.0%	40.0%	30.0%	70.0%	
	ILC	*n*	0	28	2	30	11	12	7	30	20	10	30	
		%	0.0%	28.0%	2.0%	30.0%	11.0%	12.0%	7.0%	30.0%	20.0%	10.0%	30.0%	
ER Status	Negative	*n*	1	4	6	11	1	2	8	11	4	7	11	0.02 *
		%	1.0%	4.0%	6.0%	11.0%	1.0%	2.0%	8.0%	11.0%	4.0%	7.0%	11.0%	
	Positive	*n*	11	58	20	89	27	45	17	89	56	33	89	
		%	11.0%	58.0%	20.0%	89.0%	27.0%	45.0%	17.0%	89.0%	56.0%	33.0%	89.0%	
PR Status	Negative	*n*	2	15	13	30	5	11	14	30	15	15	30	0.413
		%	2.0%	15.0%	13.0%	30.0%	5.0%	11.0%	14.0%	30.0%	15.0%	15.0%	30.0%	
	Positive	*n*	10	47	13	70	23	36	11	70	45	25	70	
		%	10.0%	47.0%	13.0%	70.0%	23.0%	36.0%	11.0%	70.0%	45.0%	25.0%	70.0%	
HER2 Status	Negative	*n*	12	58	23	93	26	45	22	93	58	35	93	0.373
		%	12.0%	58.0%	23.0%	93.0%	26.0%	45.0%	22.0%	93.0%	58.0%	35.0%	93.0%	
	Positive	*n*	0	4	3	7	2	2	3	7	2	5	7	
		%	0.0%	4.0%	3.0%	7.0%	2.0%	2.0%	3.0%	7.0%	2.0%	5.0%	7.0%	
Ki-67	Mean 20.06%		4.83	16.74	33.81		10.89	18.87	32.56		11.32	33.18		0.01 *

## Data Availability

All the data can be found in the article.

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
