# Peer review of "On the Additional Information Provided by 3T-MRI ADC in Predicting Tumor Cellularity and Microscopic Behavior"

_cancers, 2021, doi:10.3390/cancers13205167_

Round 1

Reviewer 1 Report

The approach of the study appears original. The contents of the manuscript are quite interesting by his methodology and through the tools of quantification used. I find it interesting. I thus find that this paper definitively delivers results that will surely be of interest to the readership of the journal Cancers.

However, I have some concerns about several limitations of the study. The Authors did not perform the comparisons between cellularity with the biopsy.Thus mabey the results are a little bit missleading.

In addition, the number of IDCs was much higher than ILCs.

Author Response

Response to Reviewer 1

Comments: The approach of the study appears original. The contents of the manuscript are quite interesting by his methodology and through the tools of quantification used. I find it interesting. I thus find that this paper definitively delivers results that will surely be of interest to the readership of the journal Cancers.

However, I have some concerns about several limitations of the study. The Authors did not perform the comparisons between cellularity with the biopsy. Thus mabey the results are a little bit missleading.

Thank you to the reviewer for this suggestion which allows us to better depict the analysis that has been made both for biopsy and surgical cellularity. In particular we reported in the text: “A high correlation was found between ADC values and cellularity rate, both biopsy (p<0.01) and surgical (p<<0.001)” …. For biopsy: “There was a significant difference in ADC values when the biopsy cellularity groups were compared individually, with a p value of 0.014 for <50% vs >70%. No significant difference was found for the other classes (<50% vs 50-70% and 50-70% vs >70%)”…. And for cellularity: “ADC values were also statistically different for the comparison of the individual surgical cellularity groups, with a p value <<0.001 for <50% vs >50-70%, whereas p was 0.07 for 50-70% vs >70% and p <<0.001 for <50% vs >70%”. In the last part of discussion, these sentences have been also added: “The major contribution of our work lies in demonstrating that ADC values predict main macroscopic cell growth and replication factors in a highly accurate manner. In particular, our results reinforce the current literature regarding grading and ki67, and introduce a new chapter regarding cellularity. There are no works that compare cellularity in the biopsy and in the surgical phases, therefore this work needs further validation. The ability of the ADC to predict surgical cellularity before surgery, during the MRI staging phase, certainly appears promising, offering an additional quantitative tool for the evaluation of the entire tumor, assisting the information obtained from the biopsy, which represents only tumor portions”

In addition, the number of IDCs was much higher than ILCs.

Thanks to the reviewer observation. This aspect has been reported as limitation in the last part of discussion paragraph: “another limitation is the presence of both IDCs and ILCs in the study population: the significantly lower number of ILCs may have contributed to bias in the statistics”. Evenmore we added another sentence to make it clearer: “However, the study aims to analyze the specific macroscopic histological indicators of aggressiveness and suggestive of cell growth and replication such as grading, Ki-67 and cellularity rate.”

Reviewer 2 Report

This is a retrospective analysis comparing ADC with various pathological characteristics of breast cancer patients. 

Major issues:

The authors analysis points out at least 4 other reviews on the topic.

They observe: Our results regarding the correlation between ADC values and Ki-67 are in line with what has been demonstrated in the literature, where an inverse correlation between ADC and Ki-67 index has already been shown [40-43].

Moreover, they state: The associations and correlations of ADC values with tumor subtypes, and particu- 342
larly with molecular predictive factors, are highly variable between studies, making the 343
application of DWI as a marker of aggressiveness not precise and therefore still worthy of 344
study. Some papers did not find a significant association between ADC values and histo- 345
logical grade of breast cancer [34, 35] and there is currently no consensus on the b values 346
to determine ADC values, and no cut-off has yet been proved to predict pre-biopsy tumor 347
grade.

However, it is unclear that the current analysis effectively addressed either of those problems.

Minor: 

  1. The introduction should be more succint.
  2. Many sentences are too wordy. "Nowadays" MRI is "always" used, etc.
    1. as an example the following would be better as 2 sentences:  
      1. The biopsy is the first step for the histological evaluation of the breast tumor which, however, can only be partially representative of the entire tumor.

Author Response

Response to Reviewer 2

Comments This is a retrospective analysis comparing ADC with various pathological characteristics of breast cancer patients. 

Major issues: The authors analysis points out at least 4 other reviews on the topic.

They observe: Our results regarding the correlation between ADC values and Ki-67 are in line with what has been demonstrated in the literature, where an inverse correlation between ADC and Ki-67 index has already been shown [40-43].

Moreover, they state: The associations and correlations of ADC values with tumor subtypes, and particu-342 larly with molecular predictive factors, are highly variable between studies, making the 343 application of DWI as a marker of aggressiveness not precise and therefore still worthy of 344 study. Some papers did not find a significant association between ADC values and histo- 345 logical grade of breast cancer [34, 35] and there is currently no consensus on the b values 346 to determine ADC values, and no cut-off has yet been proved to predict pre-biopsy tumor 347 grade. 

However, it is unclear that the current analysis effectively addressed either of those problems.

Thanks to the reviewer’s comment we can explain this point, maybe confused in the text.  Indeed, the “molecular predictive factors” meant the receptor structure of the cell surface (eg ER, PgR) and not the Ki67. However the sentence generated misunderstanding for which it was eliminated.  Furthermore, some other phrases that might seem contradictory have been modified.  The paragraphs concerning the analysis of the three main points (grading, ki67 and cellularity) has been reorganized to make easier the reading and highlight our contribute to literature for each of these points.  Also the last part of discussion was reinforced with our main contribution.

Minor:

1. The introduction should be more succint.

Thanks to the reviewer suggestion. The introduction has been modified and summarized

2. Many sentences are too wordy. "Nowadays" MRI is "always" used, etc.

This contradiction has been removed (“MRI is the imaging technique routinely used”) and the whole text was analyzed based on the reviewer indication

1. as an example the following would be better as 2 sentences:

1. The biopsy is the first step for the histological evaluation of the breast tumor which, however, can only be partially representative of the entire tumor.

As suggested, the sentence has been split (“Biopsy is the first step in the histological evaluation of breast cancer. However, biopsy may only be partially representative of the entire tumor”).

Round 2

Reviewer 2 Report

The manuscript is much improved. The authors have addressed my concerns. 

Author Response

Thanks to the Reviewer for helping us to improve our article.